# Unveiling the Secrets of $^1$H-NMR Spectroscopy: A Novel Approach Utilizing Attention Mechanisms

**Oliver Schilter**
IBM Research
Säumerstrasse 8
Rüschlikon, ZH 8803
oli@zurich.ibm.com

**Marvin Alberts**
IBM Research
Säumerstrasse 8
Rüschlikon, ZH 8803

**Federico Zipoli**
IBM Research
Säumerstrasse 8
Rüschlikon, ZH 8803

**Alain Vaucher**
IBM Research
Säumerstrasse 8
Rüschlikon, ZH 8803

**Philippe Schwaller**
EPFL Lausanne
Rte Cantonale
Lausanne, VD 1015

**Teodoro Laino**
IBM Research
Säumerstrasse 8
Rüschlikon, ZH 8803

## Abstract

The significance of Nuclear Magnetic Resonance (NMR) spectroscopy in organic synthesis cannot be overstated, as it plays a pivotal role in deducing chemical structures from experimental data. While machine learning has predominantly been employed for predictive purposes in the analysis of spectral data, our study introduces a novel application of a transformer-based model's attention weights to unravel the underlying "language" that correlates spectral peaks with their corresponding atom in the chemical structures. This attention mapping technique proves beneficial for comprehending spectra, enabling accurate assignment of spectra to the respective molecules. Our approach consistently achieves correct assignment of $^1$H-NMR experimental spectra to the respective molecules in a reaction, with an accuracy exceeding 95%. Furthermore, it consistently associates peaks with the correct atoms in the molecule, achieving a remarkable peak-to-atom match rate of 71% for exact match and 89% of close shift matching ($\pm$ 0.59ppm). This framework exemplifies the capability of harnessing the attention mechanism within transformer models to unveil the intricacies of spectroscopic data. Importantly, this approach can readily be extended to other types of spectra, showcasing its versatility and potential for broader applications in the field.

## Introduction

Determining the success of a chemical reaction represents a formidable challenge in experimental chemistry. It involves the purification of the reaction mixture followed by a meticulous analysis to verify the presence or absence of the desired product. Among the plethora of spectral techniques at the disposal of organic chemists, such as mass spectrometry [1, 2] and infrared spectroscopy[3, 4, 5, 6], one of the most prominent and indispensable tools is $^1$H-NMR spectroscopy[7, 6]. NMR spectroscopy operates by exposing nuclei, such as hydrogen nuclei (protons), to a magnetic field and radiofrequency pulses. These nuclei then resonate at specific frequencies, providing valuable information about a compound's structure and environment[8, 9]. From the shifts and shapes (Multiplet Type) and peak area observed in NMR spectra, a skilled chemist can decipher crucial structural information, such as the connectivity of atoms and their spatial arrangement within a molecule, thus unraveling the compound's chemical identity[10].

NeurIPS 2021 AI for Science Workshop.

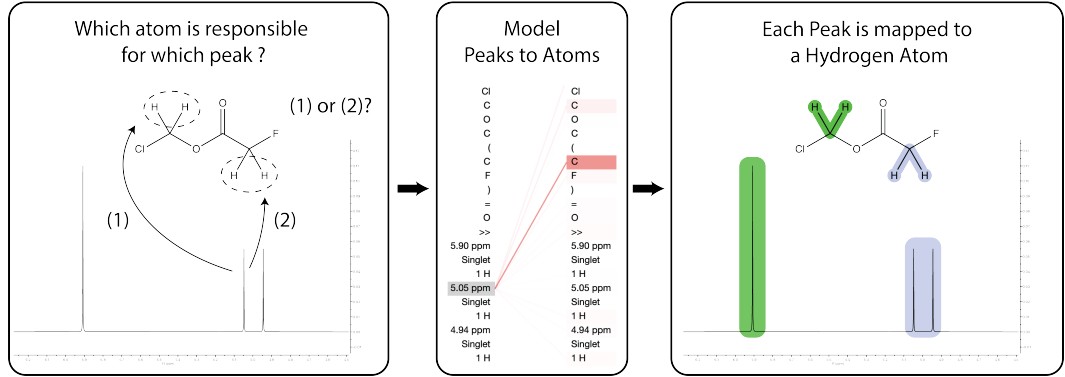

Figure 1: Assigning a peak to an atom within a molecule can be a challenging endeavor. Here, we introduce a novel attention-guided mapping strategy that facilitates a precise and clear correspondence between atoms and peaks.

As molecules can be effectively represented using string notations such as simplified molecular-input line-entry system (SMILES) [11], the field of chemistry has witnessed the integration of Natural Language Processing (NLP) techniques [12] to address various tasks, including molecular property prediction [13, 14], yield prediction [15, 16, 17], and predicting the outcomes of chemical reactions [18, 19, 20, 21]. While in the realm of spectral analysis, machine learning initially found application in predictive tasks, such as predicting a spectrum for a given molecule [22, 23, 24], a recent paradigm shift has emerged. This shift involves transitioning from the spectral domain to the molecular structure, with efforts focused on learning to decode molecular structure directly from spectroscopic data [25, 26, 27]. These models are usually directly trained to solve a specific task as an objective. In contrast, Schwaller et al.[28] showed that the attention weights of Bidirectional Encoder Representations from Transformers (BERT[29]) models learn the underlying structure of chemical reactions without any human labeling or supervision. They showcased that the self-attention mechanism and the resulting attention weights mapped the atom from the products to the reactants with an accuracy over 99.94%. So-called atom-mapping is an NP-hard problem for classical algorithms.

Here we present compelling evidence that a similar attention-weight mapping strategy can effectively correlate the positions of atoms in a SMILES string with the corresponding peaks in an [1]H-NMR spectrum (see Figure 1). We accomplished this by training BERT-based models in an unsupervised, unlabeled manner, using masked language modeling (MLM) on tokenized SMILES strings and their corresponding [1]H-NMR spectra. These models were trained using synthetic and experimental [1]H-NMR spectra, achieving a commendable accuracy rate of 71% for precise peak-to-atom mapping and over 89% accuracy for mapping within a close spectral range (within $\pm$ 0.59 ppm). Furthermore, our models demonstrate the ability to accurately identify the correct molecule from a list of candidate molecules, achieving an accuracy exceeding 95%. This promising approach holds the potential for extension to various other types of spectra, offering valuable insights into their underlying structures and paving the way for new avenues in spectroscopic analysis.

## Attention-guided [1]H-NMR mapping

Self-attention is one of the key factors in the recent success of transformer architectures such as GPT-4[30] and BERT[29]. BERT models utilize self-attention mechanisms across multiple layers to acquire a contextual representation for each token by considering all the tokens present within the same input. Every layer consists of several self-attention modules referred to as "heads", each of which is responsible for independently learning to pay attention to input tokens. When Transformers are applied in the context of molecules and their spectra, the attention mechanism is employed to focus on the atoms essential for comprehending molecular structures and elucidating the shift, shape and integral of the peaks in the spectrum. These atom representations have the potential to encapsulate a wealth of information that would be challenging for a human expert to manually capture. Thankfully, the internal attention mechanisms are readily understandable and interpretable through

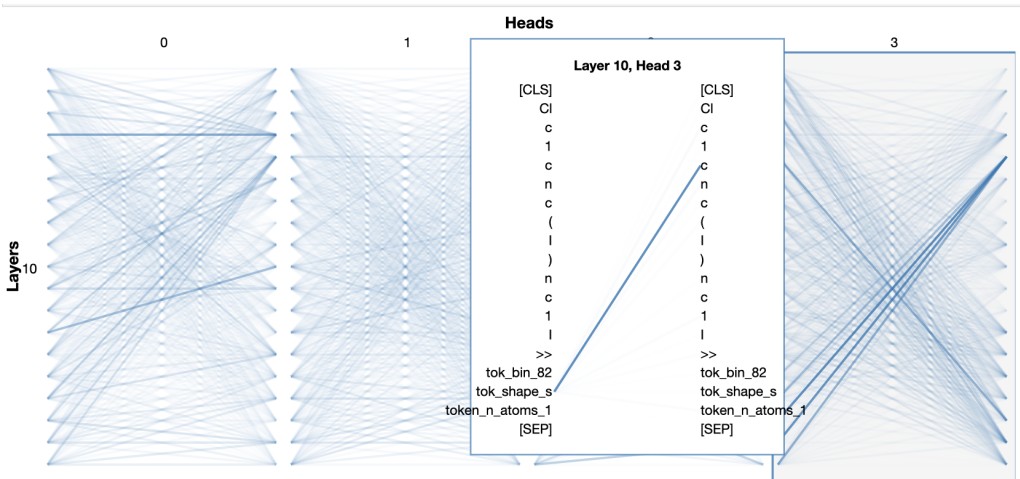

Figure 2: Explorative visual inspection of attention heads revealed that the 10 layer and 3 head showed a promising link between the correct carbon atom token to the peak token of the[1]H-NMR spectrum.

interactive visual tools[31]. With the aim of constructing such a model, we decided to train a BERT (for detailed information, refer to Section A.2). Due to the scope of this paper, we focus further discussion on the best-performing model configuration and training strategy. This model was trained using an MLM approach with the input consisting of a tokenized molecule, along with three key features for each peak in the NMR-spectrum: the peak shift (representing the centroid of the peak in ppm), the multiplet of the peak (indicating the shape of the peak), and the integral of the peak (reflecting the number of corresponding hydrogen atoms). We initially employed a set of simulated [1]H-NMR spectra from Alberts et al. [27] and subsequently fine-tuned the model using experimental [1]H-NMR spectra extracted from patent data (see Section A.1 for more information about the data and tokenization strategy). Upon conducting an initial visual assessment (see Figure 2), we pinpointed attention heads that showed potential.

## Assigning Peaks to Atoms

This discovery served as the basis for developing an algorithm with the purpose of establishing a link between attention weight matrices and connecting peak tokens with atom tokens (see Figure 3). The algorithm takes an attention matrix (for a given head and layer) and a set of tokens, distinguishing between SMILES and spectra portions. It iteratively assigns peaks to atoms by calculating attention-based probabilities, ensuring that each peak is assigned to the most appropriate atoms based on hydrogen counts and attention weights (refer to Section A.3 for more details).

To assess the algorithm's capacity for accurately mapping peaks to their respective atoms within a molecule, we utilized ChemDraw-generated [1]H-NMR Spectra, which, unlike the experimental and synthetic datasets, allowed us to extract the mapping of each chemical shift to its associated atom. We then computed the peak-to-atom accuracy for each peak and averaged it across all peaks. It's worth noting that because certain peaks in the[1]H-NMR spectrum are closely situated, we also conducted a proximity-based analysis. This involved checking whether the ML-model-assigned token received a correct prediction when it fell within a range of $\pm$ 0.59 ppm respectively within $\pm$ 5% of the covered chemical shift space. These metrics were employed to systematically evaluate the spectra mapping proficiency of all layer and head combinations within our models, replacing the need for manual visual inspection. For our top-performing BERT model, we observed that the fourth attention head in the 11th layer exhibited the highest overall performance. It achieved a peak-to-atom accuracy of 71.59% for the standard analysis and an 89.11% for proximity-based analysis. Notably, we observed a consistent trend where larger molecules tended to result in more mismatches, which can be attributed to a negative correlation between the input length and accuracy. In addition to these findings, several other layer and head combinations also demonstrated notable capabilities in peak-to-atom mapping.

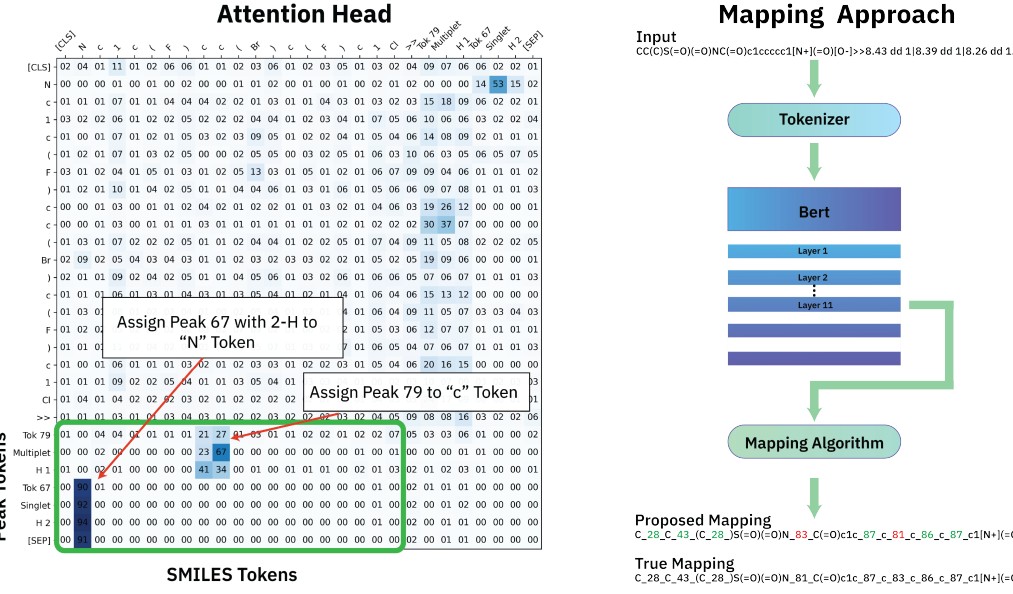

Figure 3: (left) Visualized here is the attention matrix of the performing attention head (head 3 of Layer 11). The area highlighted in green represents the mapping region, where the x-axis corresponds to SMILES tokens and the y-axis to peak tokens. Any regions outside of this green area are discarded when assigning peaks and atoms. The asymmetry in the attention matrix is caused by the self-attention mechanism underlying BERT models, where each token's attention to others is determined independently during training, allowing the model to capture diverse and contextually relevant relationships.(right) Summary of the mapping approach: Once the SMILES-Spectra input has been tokenized, the attention matrix is fed into our Mapping Algorithm. This algorithm produces a mapped SMILES representation, which serves as the basis for calculating peak-to-atom accuracy.

For instance, in layer 12, head 3 achieved an peak-to-atom accuracy of 62%, while in layer 10, head 3 exhibited a 60% peak-to-atom accuracy.

## Matching Spectrum to SMILES

The primary purpose of our model is to assist chemists in interpreting recorded spectra and proposed molecular structures. For instance, consider a scenario where a chemist records a [1]H-NMR spectrum following a chemical reaction. While the chemist knows the expected product of the reaction, uncertainty exists regarding whether the reaction indeed occurred or if the measured substance remains the starting material. The confidence level derived from the attention mapping can help determine whether a given [1]H-NMR spectrum corresponds to a specific molecule. To evaluate this capability, we assessed the model's predictions by combining reactants-SMILES, reagents-SMILES, and product-SMILES with the product's spectra for each reaction. If the model assigned the highest confidence value to the product's SMILES, we categorized the reaction as correctly predicted. Conversely, if a precursor exhibited a higher confidence value, we labeled the reaction as incorrectly predicted. The confidence score is computed by averaging the attention weights associated with the peaks mapped using the H-Mapping algorithm, while also factoring in ratios that penalize inconsistencies in hydrogen atom-to-peak mapping and deviations from an equal number of hydrogen in the SMILES notation versus numbers of hydrogen in the spectra (see Section A.4).

We conducted a comprehensive evaluation of the entire experimental dataset for this task, achieving an accuracy rate of 85.81%. This success is notably higher than the random baseline accuracy of 20.95%, which essentially amounts to randomly assigning correct spectra to molecule candidates. Upon closer examination of misclassified reactions and spectra, we observed that discrepancies in the number of hydrogen atoms between the spectra and the product molecules had a significant impact on confidence scores. This prompted us to omit instances where the sum of hydrogen atoms deviated from that of the corresponding molecules, as such disparities stand as indicators of potential errors in

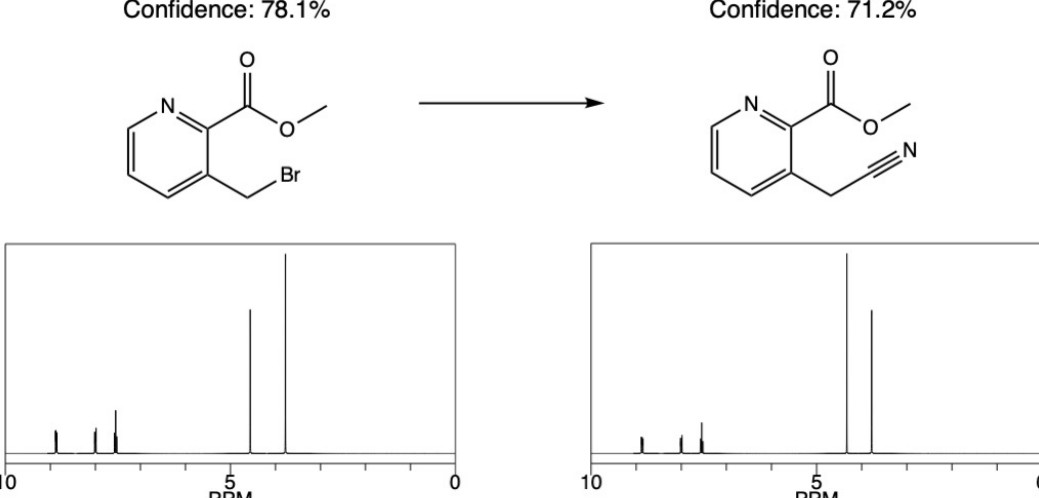

Figure 4: The confidence score can fail to assign a spectrum to the correct molecule if the H-NMR spectrum of the reactant (left) is close to the products spectrum (right) this results in the predicted confidence being higher for the reactant, causing a misclassification.

the spectral extractions or inaccuracies in the spectra themselves. Upon excluding cases where the number of hydrogen atoms in the spectra differed from the sum of hydrogen atoms in the product molecules, our accuracy rate increased to 95.10%. This increase in accuracy is explained by the fact that if the number of hydrogen atoms in the wrong spectra closely aligns with the hydrogen count of one reactant instead of the hydrogen count of the product, the confidence tends to favor that reactant over the product.

Our qualitative analysis revealed that our approach faced its greatest limitations when dealing with reactions where the changes between the reactants were smaller in terms of affected hydrogen-carrying atoms. An illustrative example can be found in Figure 4. On average, when misclassifications occurred, the reactant with the highest confidence was 98.8% the length of the product, compared to an average length difference of 82.9% between the product and reactants. This demonstrates that misclassifications primarily occurred in challenging edge cases with similar spectra, making them difficult even for human observers to distinguish.

Overall our method allows reliable assignments of spectras to the correct molecular structure on real-world spectras. The insights that the mapping provides give the chemist in charge a better understanding and help in the structure elucidation.

## Conclusion

We have demonstrated that self-supervised attention-based transformer models can learn the correlation between [1]H-NMR peaks and the atoms bearing the corresponding hydrogen responsible for the peak. Through careful experimentation, we identified the optimal combination of attention layer and head, resulting in the most robust attention-based connection between the peak and its corresponding atom. Leveraging this insight, we applied our [1]H-mapping algorithm to consistently and accurately link peaks to their respective atoms, all without the need for expert encoding. Furthermore, we showcased the practicality of our approach by successfully deciphering real-world spectra, and reliably associating them with the correct molecules. A key advantage of our methodology that could be explored in the future is its adaptability to various spectra, potentially offering a versatile tool for extracting valuable insights from diverse spectroscopic data.

# A Appendix

## A.1 Data

The three datasets evaluated in this study are based on the Pistachio dataset obtained from NextMove [32]. This source, derived from reactions extracted from patent data. Subsequently, synthetic data was generated based on the reagents, reactants, and products found within the pistachio dataset, by Alberts et al. in their work [27]. Their synthetic dataset was created using the [1]H-NMR prediction function within MNova[33], resulting in what we refer to as the "simulated-dataset". The exact procedure and molecule selection criterion's can be found in literature as well as on GitHub[1].

Experimental data, on the other hand, was extracted from paragraphs within the Pistachio dataset using a set of regular expressions to capture peak information. This extracted data was further curated (e.g. dropping entries with parsing errors in the spectra) with and is denoted as the "experimental-dataset". Additionally, a smaller subset of data was simulated using ChemDraw, adhering to standard settings but modifying the solvent to deuterated chloroform, aligning with the approach outlined by Alberts et al. [27]. This subset is referred to as the "Mapped-dataset". This is the case since ChemDraw generated[1]H-NMR Spectrum allow the mapping of the peak position directly to the atom in the molecule respectively the token position in the SMILES to peak in the spectra. A summary of dataset sizes and their respective scopes can be found in Table 1.

Each entry in the dataset consists of a SMILES string followed by a separating character '»', and subsequently, a list of shifts, shapes, and H-integral values. These values are organized in descending order of peak shift, following the format:
*SMILES » Shift1 + Shape1 + Integral1 | Shift2 + Shape2 + Integral2*.

All the data underwent tokenization using a modified SMILES tokenizer, as outlined in [34]. This tokenizer was adapted not only for SMILES but also to handle tokens for peak shape (multiplet), peak shift and peak integral values. The continuous chemical shift data was discretized into 100 bins, tailored to the shift range represented in the datasets, while for the the shape and integral discrete tokens for each possibility were extracted. The tokenizer's vocabulary was constructed based on the synthetic data and was consistently applied across all experiments. In cases where an entry from another dataset lacked supported tokens, it was excluded from consideration, rather than relying on an unknown token for processing. For the data splitting, we followed the practice outlined in the literature [27], allocating 85% for training, 5% for validation, and 10% for testing purposes. [27].

Table 1: Size of a Data Sets

|  | Train | Validation | Test |
| --- | --- | --- | --- |
| Simulated-Dataset | 2'690'762 | 141'620 | 314'710 |
| Experimental-Dataset | 1'123'107 | 59'079 | 131'171 |
| Mapped-Dataset |  |  | 100 |

## A.2 Machine Learning Architecture

Our implementation drew inspiration from the Reaction Mapper models introduced by Schwaller et al. [28, 34], incorporating both ALBERT [35] and BERT [29]. In order to acquaint our models with the underlying "language" of the spectra, we chose a Mask Language Modeling task for training. In this task, 15% of the tokens were randomly masked, and a cross-entropy loss was applied to the masked token positions, enabling the models to learn token recovery. The architecture of the models was inspired by Schwaller et al. [28, 34]: Layers 12, Heads 4, Hidden size 256, Hidden dropout 0.1, Hidden activation "GELU", Batch size 16 and Learning rate 0.00002. We explored three preliminary BERT models during our investigation: (1) trained exclusively on synthetic data, (2) trained solely on experimental data, and (3) pretrained on synthetic data before fine-tuning on experimental data. Model (1) and (2) underwent 48 hours of training on an Nvidia A100, while for Model (3), we initialized training with the checkpoint from Model 1 after 24 hours and continued for an additional 24 hours. The (3) model was trained for 2.9mio steps (2x15epochs) and overall performed the best in preliminary studies achieving the highest peak-to-atom accuracy and therefore used in this study.

---

[1]https://github.com/rxn4chemistry/nmr-to-structure

### A.3 H-mapping Algorithm from Attention

1. **Input and Preparation:**
   The algorithm takes as input an attention matrix (n x n) corresponding to a specific layer and head for a given set of n tokens. These tokens contains both the SMILES and spectra tokens, distinguished by a special token, allowing for the reconstruction of the original SMILES. Using RDKit, we determine the number of hydrogens for each (atom) token in the SMILES.

2. **Masking the Attention Matrix:**
   Next, we mask the attention matrix to exclude tokens that do not carry hydrogen atoms in the SMILES as well as other special tokens such as the separating, starting and end-of-sentence tokens.

3. **Iterating through Peaks:**
   For each peak the number of hydrogen is assigned taken from the corresponding integral token. Then all combinations of H-carrying atoms are generated, but only the combinations which sum of hydrogen atoms is equal or smaller then the integral (H-number of the peak) are kept. A combination can also be a single atom. For each possible combinations the attention weights are taken from the matrix (shift position of the peak for row, and column(s) for the atom(s). If a combination is corresponding to multiple atoms, the attention is averaged. This attention is then multiplied by the ratio of sum of hydrogen from the atoms divided by peak number of hydrogens. This is done to punish possible combinations that not covert the full integral of the peak. Then a table with the peak, attention and the corresponding atoms is generated over all peaks.

4. **Peak Assignment:**
   We sort the table for the highest attention value. We then iterate trough the table, assign the combination with the highest attention value, assign the peak to the corresponding atom(s). Then all entries including the assigned atom(s) and peak are removed from the table. We iterated trough the table until the table is empty hence all peaks are assigned to atoms.

### A.4 H-mapping Confidence Score

The confidence score was computed as follows: After applying the H-Mapping algorithm the attention weight of each mapped peak was averaged and adjusted by two key ratios: The first ratio accounted for the relationship between mapped peaks and atoms. It was designed to penalize instances where a hydrogen atom was not mapped to a peak, ensuring that confidence was diminished accordingly. The second ratio considered the ratio of hydrogens (or the sum of peak integrals) over the total number of hydrogens in the SMILES. Importantly, these ratios were calculated to always place the larger number in the denominator. This approach effectively penalized any deviations from an equal ratio, providing a robust measure of confidence.

## Acknowledgments and Disclosure of Funding

This publication was created as part of NCCR Catalysis (grant number 180544), a National Centre of Competence in Research funded by the Swiss National Science Foundation.

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
