# OpenReview forum: "Unveiling the Secrets of $^1$H-NMR Spectroscopy: A Novel Approach Utilizing Attention Mechanisms"
_NeurIPS.cc/2023/Workshop/AI4Science — NeurIPS2023-AI4Science Poster_

### Official Review · Reviewer_Ypab · 2023-10-06
**A good example of experimental 1H-NMR data analysis with ML**

**Rating:** 8
**Confidence:** 3

**Review:**

## Summary

The authors train a BERT-based models in an unsupervised, unlabeled manner, using masked language modeling (MLM) on molecules represented as tokenized SMILES strings and the molecules corresponding 1H-NMR spectra. These models achieve a commendable accuracy rate of 71 % for precise peak-to-atom mapping and over 89 % accuracy for mapping within a close spectral range (within ±0.59ppm). Their models also demonstrate the ability to identify the correct molecule from a list of candidate molecules with 95+ % accuracy.

## Evaluation of the quality

The authors presents a novel ML model that achieves convincing results for analysis of 1H-NMR spectra. The authors train the model on simulated data but fine-tune it on experimental data. They also evaluate it on experimental data with good results. Therefore, I nominate this paper for acceptance with a high score (7-10).

## Clarity

The abstract, introduction and conclusion are well-written. The remainder of the paper suffers from issues of readability that could be significantly improved. Specifically, enhanced guidance for the reader such as introducing figures and their relevance before they appear could make a substantial difference.

It is not always clear what the accuracies refer to: "for example, the 71 % and 89 % in the abstract"
Please make it clear for every reported accuracy whether it is on simulated, experimental or mapped dataset. For example make a large table with all of these accuracies which can be referred to throughout the paper.

## Originality and significance

The work is very significant as it can be used to analysis experimental 1H-NMR data and is not restricted to simulated data.

I hope the authors can use my comments to improve the papers.

Best of luck :)

---

### Official Review · Reviewer_Fk3Q · 2023-10-25

**Rating:** 4
**Confidence:** 3

**Review:**

Summary:

This work describes a method of matching H-NMR peaks and the atoms with hydrogen which are responsible for those respective peaks. The attention matrix from a specific layer/ attention head from a transformer based-model is used to predict the correlation

Comments:

1. A positive aspect of this work is that it is not solely based on synthetic data. Real experimental data are also used for training and validation.

2. The main contribution of this work seems to rely on selecting a specific layer and attention head before applying the H-mapping algorithm. This selection also seems to be manual based on visual inspection. This appears to be a non-trivial bottle neck towards application of this algorithm for any potential peak assignment pipeline.

3. The left panel of figure 3 shows an attention matrix. It is a square matrix with the same set of entries in the X and Y axes. I am wondering in this case why we don’t have symmetric entries. It will be helpful to add a bit more details on this for readers who are not too familiar with this topic to understand this discrepancy.

Though the premise of this work is important in the field of chemistry, the actual contribution detailed in this paper is not of significant impact.

---

### Meta-Review · Area_Chair_HPNn · 2023-10-26

**Recommendation:** Accept (Poster)
**Confidence:** 4

**Metareview:**

The paper explores a timely topic of automatic processing of H-NMR spectra. This indeed becomes a pressing problem with the rise of automation of chemistry laboratories. The proposed approach to use Transformer's attention is novel and simple. One doubt it raises is that it is somewhat hard to iterate on the approach: it is not obvious what contributes to the successes and failures of the approach, as the attention strengths are not directly supervised to align the spectra. One reviewer remarked that the attention layer was selected manually, which indeed if is the case, is a weakness (that I suppose can be fixed by optimizing the layer choice on a held out validation set). Overall the work while arguably preliminary, is interesting and of value to the community. Hence, it is my pleasure to recommend the acceptance of the paper.